# Immune activation of Bio-Germanium in a randomized, double-blind, placebo-controlled clinical trial with 130 human subjects: Therapeutic opportunities from new insights

**Jung Min Cho[1,2], Jisuk Chae[1,2], Sa Rang Jeong[1,2], Min Jung Moon[1,2], Dong Yeob Shin[3], Jong Ho Lee**[1,2]*

1 National Leading Research Laboratory of Clinical Nutrigenetics/Nutrigenomics, Department of Food and Nutrition, College of Human Ecology, Yonsei University, Seoul, Republic of Korea, 2 Department of Food and Nutrition, College of Human Ecology, Yonsei University, Seoul, Republic of Korea, 3 Division of Endocrinology and Metabolism, Department of Internal Medicine, Yonsei University College of Medicine, Seoul, Republic of Korea

* jhleeb@yonsei.ac.kr

**Data Availability Statement:** All relevant data are within the manuscript.

## Abstract

Germanium has long been considered a therapeutic agent with anticancer, antitumor, anti-aging, antiviral and anti-inflammatory effects. Numerous clinical studies have explored the promising therapeutic effects of organic germanium on cancer, arthritis and senile osteoporosis. The immune activation property of organic germanium is considered the foundation of its various therapeutic effects. However, previous human clinical studies investigating immune activation with organic germanium compounds have certain limitations, as some studies did not strictly follow a randomized, double-blind, placebo-controlled design. To build a more clinically substantiated foundation for the mechanism underlying its immunostimulation, we structured by far the most rigorous clinical study to-date with a group of 130 human subjects to examine changes in immune profiles following germanium supplementation. We used Bio-Germanium, an organic germanium compound naturally synthesized *via* a yeast fermentation process. An 8-week randomized, double-blind, placebo-controlled study was conducted with 130 subjects with leukocyte counts of 4–8 ($\times10^3$/μL) divided into the Bio-Germanium group and the placebo group. Anthropometric measurements; blood collection; biochemical analysis; urinalysis; and natural killer cell activity, cytokine and immunoglobulin assays were conducted. Results showed the Bio-Germanium group exhibited NK cell activity increases at effector cell:target cell (E:T) ratios of 50:1, 10:1, 5:1 and 2.5:1 (12.60±32.91%, 10.19±23.88%, 9.28±16.49% and 7.27±15.28%, respectively), but the placebo group showed decreases ($P<0.01$). The difference in the IgG1 change from baseline to follow-up between the Bio-Germanium and placebo groups was significant ($P = 0.044$). Our results and earlier clinical study of Bio-Germanium confirm that Bio-Germanium acts as an effective immunostimulant by increasing the cytotoxicity of NK cells and activating immunoglobulin, B cells and tumor necrosis factor (TNF)-α ($P<0.05$). As we have added newly discovered clinical findings for germanium's immunostimulation mechanism, we

**Funding:** This work was supported by the New Drug Discovery Fund of Geranti Pharmaceutical. The funders had no role in study design, data collection and analysis, decision to publish, or preparation of the manuscript.

**Competing interests:** This work was supported by the New Drug Discovery Fund of Geranti Pharmaceutical and was supervised by a contract research organization (CRO, NeoNutra Co., Ltd). This financial support does not alter our adherence to PLOS ONE's policies concerning sharing data and materials. The funder had no role in the study design, data collection and analysis, decision to publish, or preparation of the manuscript. In particular, the funder never invited, offered or guaranteed opportunities in employment, consultancy, patent registration, product development or any other related capacities to any of the authors of this study. The CRO supervised and managed with strict vigilance all stages of the clinical trial process to ensure that the laws, regulations and international standards designed for data integrity were maintained.

believe Bio-Germanium is a highly promising therapeutic agent and should certainly be further explored for potential development opportunities in immunotherapy.

## Registered clinical trial

[NCT03677921]; www.clinicaltrials.gov
[KCT0002726]; https://cris.nih.go.kr

## Introduction

Germanium is a naturally occurring ultratrace element with a wide application range, from the electronics industry to the dietary supplement industry [1, 2]. Organic germanium has been reported to be a therapeutic agent with anticancer [3], antitumor [4, 5], antiaging [6, 7], antiviral [8] and anti-inflammatory [9, 10] effects; its anticancer and antiviral effects have been observed *in vivo*, and its antitumor, antiaging and anti-inflammatory effects have been observed both *in vivo* and *in vitro*. In addition, organic germanium compounds have been effective in treating cancer and arthritis [11, 12] and in enhancing immune function [13–15] in pathological conditions in preclinical studies.

Numerous clinical studies have explored the promising therapeutic effects of organic germanium on disease. Human clinical trials in lung cancer have revealed that organic germanium supplementation benefited survival, tumor regression and overall improvements in performance status and immunological parameters [16]. Other clinical investigations have demonstrated the preventive and therapeutic effects of organic germanium on senile osteoporosis through the enhancement of osteoblast activity [17]. In cancer patients especially, organic germanium ameliorated side effects from chemotherapy and radiotherapy [18]; lowered the rates of cancer metastasis and recurrence [18]; prolonged the survival and relieved the pain of terminal cancer patients [19]; and resulted in partial and complete remission of cancer [20, 21].

Generally, the immune activation property of organic germanium is deemed to be the foundation of its various therapeutic effects, such as anticancer, antiaging, antiviral and antiinflammatory effects—areas in which immune function plays a key role. In preclinical studies, germanium's immune activation was observed in a wide spectrum of immune system functions, evidenced by the activation of NK cells, T cells, macrophages, neutrophils, lymphokines, and interferons [15, 22–25]. In clinical studies, significant immunostimulatory effects were shown by the activation of NK cells, macrophages, and neutrophils in patients taking organic germanium [26–28].

Despite the abundance of studies in the literature, we believe previous human clinical studies of immune activation with organic germanium compounds have certain limitations, as some studies did not strictly follow a randomized, double-blind, placebo-controlled design. To build a more clinically substantiated foundation for the mechanism behind its immunostimulation, we believe a large-scale human clinical study is needed to bridge the gap between preclinical and clinical results. For this reason, we structured by far the most rigorous clinical study to-date with a group of 130 human subjects to examine changes in immune profiles following 8 weeks of germanium supplementation. Additionally, instead of designing targeted, disease-specific research, we chose healthy human subjects in the ages between 20 and 75 with normal immune function to establish a more universal basis so that the immune activation mechanism of germanium would have wider application.

We used Bio-Germanium, a naturally synthesized organic germanium, for our clinical trial. Bio-Germanium was developed in the late 1980s in the search for safer and more natural organic germanium. This new type of organic germanium is formulated by a natural yeast cultivation process, containing stable organic germanium bound to yeast protein [29–31]. Bio-Germanium has been subjected to various safety [1, 32–36] and efficacy [2, 10, 37–39] analyses and has been sold on the market for over 20 years. In addition, Bio-Germanium is acknowledged as a new dietary ingredient by the US Food and Drug Administration (FDA, ID: FDA-2010-S-0665-0243) and has been approved as a functional ingredient for immune enhancement by the Korea Ministry of Food and Drug Safety (KMFDS, Functional Ingredient No. 2007–15). Furthermore, Bio-Germanium has been granted green list status by the US FDA and is exempted from the import alert of germanium products (FDA Import Alert 54–07).

Our results and earlier clinical study of Bio-Germanium [38] confirm that Bio-Germanium acts as an effective immunostimulant by increasing the cytotoxicity of NK cells and activating immunoglobulin, B cells and tumor necrosis factor (TNF)-$\alpha$ ($P<0.05$). We have added newly discovered clinical findings for germanium's immunostimulation mechanism, substantiated through a large-scale trial offering a universal basis by focusing on healthy human subjects. We believe Bio-Germanium is a highly promising therapeutic agent and should certainly be further explored for potential development opportunities in immunotherapy.

## Materials and methods

### History of germanium supplements

Germanium-containing dietary supplements became popular in the 1970s in Japan, in the mid-1980s in Great Britain, and later in other countries [40]. However, due to a lack of awareness of the danger of consuming inorganic minerals, certain fatalities occurred due to the intake of inorganic germanium [40]. This is understandable because metal-based anticancer drugs, such as cisplatin, carboplatin and oxaliplatin, started to be widely used at that time for chemotherapy, and their side effects, such as nephrotoxicity, nausea, neurotoxicity, myopathy, bone marrow suppression, renal failure and metal intoxication, were considered inevitable parts of the chemotherapy and were generally tolerated for the purpose of cancer treatment, as remission of cancer took priority over side effects. As some considered germanium to be a new candidate amongst metal-based agents for chemotherapy, the behavior of continuing long-term intake of even inorganic germanium formulation could have been influenced by this notion of acceptance of side effects in desperation for treatment, subsequently resulting in those fatalities. At any rate, the importance of an organic formulation was revisited [41], and no such incidents of taking inorganic germanium have occurred since; currently, only organic germanium is widely used. In the search for safer and more natural organic germanium, a new type of organic germanium product, Bio-Germanium, was developed *via* germanium biosynthesis utilizing a natural yeast cultivation process [42]. As yeast are known to detoxify toxic metals and inorganic elements [43] through biological assimilation [37], yeast metabolic processes are utilized to convert inorganic germanium into an organic compound. The safety of Bio-Germanium has been extensively and thoroughly tested in *in vitro*, *in vivo* and human clinical studies [1, 32–36, 38], and its efficacy has been assessed in the areas of immunostimulation, antitumor effects, anti-inflammation and others [2, 10, 37–39].

### Manufacturing bio-germanium

Bio-Germanium, our study material, was manufactured and provided by Geranti Pharmaceutical. This new type of organic germanium is formulated *via* germanium biosynthesis, utilizing a natural yeast cultivation process to enhance the biological activity and reduce the toxicity of

inorganic elements. A reported advantage of trace element-concentrated yeast is the decreased toxicity of the inorganic elements [44]. Additionally, microorganisms were found to convert inorganic germanium into self-organizing germanium *via* self-accumulation; this method was confirmed as a detoxification method [45]. Yeast, such as *Saccharomyces cerevisiae*, are known to be harmless microorganisms in the human body and to play an important role as a medium for comprehensive biological research in various fields, including molecular biology and molecular genetics [46, 47]. The strain used for Bio-Germanium production was an *S. cerevisiae* strain (Korean Collection for Type Cultures, KCTC-7904, indexed as KCTC-1199 formerly) obtained from the Korea Research Institute of Bioscience and Biotechnology Gene Bank.

## Safety of bio-germanium

Bio-Germanium has been comprehensively tested for safety, with investigations ranging from its organic structure, oral toxicity, and genotoxicity to its effects when consumed by humans. Previous *in vitro* studies have confirmed that Bio-Germanium contains only organic germanium by verifying that the germanium in Bio-Germanium is protein-bound organic germanium; the inorganic form of germanium is not present in Bio-Germanium; and the germanium in Bio-Germanium does not dissociate from yeast by dissolution in either gastric juice or water [29–31]. Genotoxicity studies, such as reverse mutation, chromosomal aberration and micronucleus tests, indicated that Bio-Germanium neither causes mutagenic activity nor possesses genotoxic potential [33]. In *in vivo* studies, acute (single, 14 days), subchronic (repeated, 13 weeks) and chronic (10 consecutive months) oral toxicity studies were conducted in both rats and beagle dogs, and Bio-Germanium was shown to be safe in animal studies at dosages of 2000, 3000 and 5000 mg/kg body weight/day [1, 32, 34, 36]. Additionally, the accumulation of Bio-Germanium, particularly in the kidneys and liver, was tested; those studies showed that Bio-Germanium does not result in germanium accumulation in these organs [35]. In human studies, organic germanium compounds are known to be well absorbed and completely excreted from the body within 72 h [48–51]. Moreover, in a previous human clinical trial of Bio-Germanium conducted in 50 subjects with an 8-week, randomized, double-blind, placebo-controlled design, Bio-Germanium did not cause any adverse effects and, particularly, did not influence liver- and kidney-related biochemical markers, such as alanine aminotransferase (ALT), aspartate aminotransferase (AST), creatinine, blood urea nitrogen (BUN), total bilirubin (TB), and alkaline phosphatase (ALP), or anemia-related biochemical markers, such as hemoglobin, hematocrit, and red blood cell count, after supplementation, reconfirming its safety for human consumption [38]. Thus, through various *in vitro*, *in vivo* and human clinical studies conducted previously, the study material, Bio-Germanium, was confirmed to be a safe organic germanium suitable for consumption.

## Efficacy of bio-germanium

Bio-Germanium has demonstrated efficacy in areas such as promoting immunostimulation, inhibiting tumor progression, and conferring anti-inflammatory effects. In a study by Lee *et al.* [38], Bio-Germanium demonstrated its immunostimulatory effect on humans through the activation of tumor necrosis factor (TNF)-α and B cells (CD19) when compared with the control group. As TNF-α is involved in the regulation of immune cells, such as in macrophage phagocytosis and T cell homeostasis, and B cells function in adaptive immunity by secreting antibodies, although more in-depth assessments of the responses of antigen-specific cytotoxic lymphocytes and antibody responses to proteins and polysaccharide antigens are warranted, Bio-Germanium seems to promote stimulation in both humoral and cell-mediated immunity.

In a test of the anticancer and antitumor activities of Bio-Germanium by Baek *et al*. [2], sarcoma-180 tumor-bearing C57BL/6 mice were fed plain yeast (*S. cerevisiae*) in the negative control group, doxorubicin (chemotherapy drug) in the positive control group, and Bio-Germanium in the test group. In this study, Bio-Germanium treatment significantly reduced the tumor weight in sarcoma-180 tumor-bearing mice, suggesting that the suppressed progression of tumors may have been the result of increased TNF-α production and effector function of NK cells. Interestingly, while the doxorubicin group showed significant body weight loss, the Bio-Germanium group did not exhibit a weight decrease. In addition to this antitumor effect, the anti-inflammatory effect of Bio-Germanium against paw edema was investigated by Lee *et al*. [10]. Male Sprague-Dawley rats (180–200 g) were injected with 100 μL of 1% carrageenan to induce paw edema. Ibuprofen (nonsteroidal anti-inflammatory drug) was used as a positive control, and Bio-Germanium was applied in different concentrations as test treatments. Ibuprofen significantly inhibited carrageenan-induced edema, and Bio-Germanium also showed comparable inhibitory effects in a dose-dependent manner. Further study showed that the anti-inflammatory activity of Bio-Germanium appears to be related to the inhibition of arachidonic acid release and prostaglandin $E_2$ ($PGE_2$) production in rat basophilic leukemia cells (RBL-2H3).

## Organic properties of bio-germanium

A number of studies were conducted to determine the organic structure and properties of Bio-Germanium, primarily prepared for the KMFDS approval application for a dietary ingredient with immune enhancement function. The organic properties of Bio-Germanium were analyzed in the following steps: (1) its germanium-binding structure was investigated to determine whether the germanium in Bio-Germanium is a protein-bound germanium by conducting ion exchange chromatography and gel filtration to compare the elution profiles of plain yeast, germanium dioxide and Bio-Germanium, and by conducting sodium dodecyl sulfate-polyacrylamide gel electrophoresis (SDS-PAGE) and N-terminal amino acid sequence analyses to identify its protein type [29]; (2) the natural formulation technology *via* germanium biosynthesis was assessed to determine whether the inorganic form of germanium is fully transformed into an organic form by conducting X-ray diffraction (XRD), nuclear magnetic resonance (NMR) and Fourier transform infrared spectroscopy (FT-IR) analyses to compare the structural formation of plain yeast, germanium dioxide and Bio-Germanium [30]; (3) the integrity of the organic conversion was investigated to ensure that no inorganic germanium is present in Bio-Germanium by following a qualitative analysis protocol utilizing the unique chemical properties of $NaBH_4$ and $GeO_2$ reactions, and NMR, ultraviolet visible spectrophotometry (UV-VIS), inductively coupled plasma atomic emission spectroscopy (ICP-AES), FT-IR and XRD analyses were conducted to detect the presence of inorganic germanium in Bio-Germanium [31]; and (4) the protein-bound germanium structure and its stability were tested to determine the possibility of dissociation in gastric juice by applying dialysis membrane tubing with a size cut-off of 1200 daltons to Bio-Germanium dissolved in simulated gastric juice and assessing the presence of inorganic germanium [30]. In conclusion, Bio-Germanium is confirmed to be a protein-bound organic germanium with a stable structure and to not have any inorganic germanium present.

## Study subjects and recruitment criteria

The current study enrolled 130 healthy subjects with leukocyte counts of 4–8 ($\times 10^3$/μL), which is considered to be in the normal range for healthy humans. We screened volunteers recruited from advertisements by the Clinical Nutrigenetics/Nutrigenomics Laboratory at Yonsei

University. Adult male and female participants aged 20 years to 75 years with leukocyte counts of 4–8 ($\times 10^3/\mu L$) were eligible for inclusion. The exclusion criteria were as follows: the consumption of drugs or dietary supplements related to immune function within two weeks prior to screening; an allergy to the study products and related substances; hypertension; diabetes; any history/presence of significant metabolic disease; any history/presence of acute or chronic infection of the liver, kidney, or gastrointestinal system; and any history/presence of any other acute or chronic disease requiring treatment.

To recruit subjects with normal immune function, we used leukocyte counts because leukocytes play important roles in immune responses and defense mechanisms [52, 53]. Normal leukocyte counts are within a range of 4 to 10 ($\times 10^3/\mu L$) [54]. Leukocyte counts of less than 4 ($\times 10^3/\mu L$) are considered abnormal, and this status is diagnosed as leukopenia [55]. Conversely, Tamakoshi *et al.* reported that the relative risk (RR) of cardiovascular disease mortality in patients with leukocyte counts of 8 to 10 ($\times 10^3/\mu L$) was more than twofold greater than that in patients with leukocyte counts of 4 to 7.9 ($\times 10^3/\mu L$) (RR = 2.18, 95% confidence interval: 1.23–3.88) [56]. In addition, many studies have revealed that an elevated leukocyte level in blood circulation and vessel walls is a predictive indicator of inflammation [57–60]. Therefore, within the normal range of 4 to 10 ($\times 10^3/\mu L$), a slightly decreased upper range of 4 to 8 ($\times 10^3/\mu L$) was defined. The fasting whole blood leukocyte counts of the volunteers were measured, and 130 participants who met the inclusion criteria were ultimately enrolled. The purpose of the study was carefully explained to all subjects, and written informed consent was obtained before participation.

## Registration of the clinical trial

The study protocol was approved on December 14th, 2017 by the Institutional Review Board of Yonsei University (IRB No. 7001988-201712-HR-322) according to the Declaration of Helsinki. Our clinical trial was first registered in the Clinical Research Information Service (CRIS), which is an online clinical trial registration system established by the Korea Centers for Disease Control and Prevention (KCDC) with support from the Korea Ministry of Health and Welfare (KMOHW) and embodied as a part of the Primary Registries in the World Health Organization (WHO) Registry Network. We registered our clinical trial in CRIS on January 19th, 2018 (https://cris.nih.go.kr—Identifier: KCT0002726). The first participant was enrolled on January 9th, 2018, and the last (130th) participant's final visit ended on June 5th, 2018. A gap of ten days between the enrollment of the first participant and the trial registration dates occurred due to logistical issues caused by earlier-than-expected inflow of screening candidates for trial participation. To increase the global exposure of our trial, despite redundancy, we also registered our trial in ClinicalTrials.gov on September 12th, 2018 (https://clinicaltrials.gov—Identifier: NCT03677921). The authors confirm that all ongoing and related trials associated with this study material are registered.

## Study design and determination of group size

An 8-week, randomized, double-blind, placebo-controlled clinical study was conducted with 130 subjects with leukocyte counts of 4–8 ($\times 10^3/\mu L$) who were divided into two groups: the Bio-Germanium (test) group and the placebo (control) group. Over the eight-week testing period, participants in the test group consumed 1200 mg of Bio-Germanium daily, while those in the control group consumed placebo in the same amount. The sample size was determined and calculated by reference to the NK cell cytotoxic activity results from another clinical trial that showed statistically significant results [61]. According to the superiority test based on that reference study, the delta (change) in the NK cell cytotoxic activity at an effector cell:target cell

(E:T) ratio of 10:1 was -0.28±10.7% (mean ± standard deviation) in the placebo group and lower than that in the test group (7.4±10.0%). Based on the difference in deltas between the test and placebo groups (7.68), the application of a 75% adjustment based on the study's relevance in clinical efficacy yielded an effect size ($d$) of 5.76. The sample size was then determined *via* a two-sample *t*-test power calculation with an effect size ($d$) of 5.76, a power of 0.8, and a level of significance ($\alpha$) of 0.05. The result indicated that a minimum of 55 subjects per group was needed, and we estimated a dropout ratio of 15%; thus, we selected 65 participants to increase the statistical power of the test.

## Study material information and compliance

In this human clinical trial, Bio-Germanium was used at a dosage of 1200 mg/day for 8 weeks in accordance with the approval from KMFDS. According to the blinded and randomized allocation schedule, the participants received Bio-Germanium or placebo in individual bottles, and both the test and placebo products were provided as capsules and were identical in packaging, appearance, color, texture, and smell. The participants were instructed to consume 4 capsules per day: 2 capsules after breakfast and 2 capsules after dinner. Hydroxypropyl methylcellulose (HPMC) capsules were used to encapsulate the test and placebo ingredients. The study material information is as follows (Table 1):

Compliance was assessed by counting the remaining capsules and food records. If more than 70% of the capsules were consumed, the compliance was considered fulfilled.

## Implementation of the clinical trial

This study was designed as a randomized, double-blind, placebo-controlled clinical trial. *Via* computer-generated block randomization lists (the randomization program of SAS version 9.4; SAS Institute, Cary, North Carolina, USA), 130 participants were randomly assigned to receive either placebo or Bio-Germanium. A third-party expert randomly coded the blinded allocation schedule and strictly controlled the double-blind conditions. The principal investigators enrolled participants and allocated subjects according to the given blinded allocation schedule. Those assigned to the test group received Bio-Germanium, and those assigned to the control group received corn starch as placebo. Corn starch was chosen as it is conventionally used as placebo in human trials involving immune experiments [62, 63]. Participants, healthcare providers, investigators, outcome assessors and data analysts were kept blinded

**Table 1. Study material information.**

| Bio-Germanium | | Placebo | |
|---|---|---|---|
| **Criteria** | **Information** | **Criteria** | **Information** |
| Capsule Material | White HPMC capsule | Capsule Material | White HPMC capsule |
| Capsule Size | Size #1 capsule (300 mg) | Capsule Size | Size #1 capsule (300 mg) |
| Capsule Filling | Powder form | Capsule Filling | Powder form |
| Dosage Guidance | 4 capsules per day (1200 mg/day) | Dosage Guidance | 4 capsules per day (1200 mg/day) |
| Main Ingredient | Bio-Germanium (100%) | Main Ingredient | Corn starch (96.7%) |
| Appearance | Powder in light brown color | Color Additive 1 | Brown (2.0%) |
| Flavor | Slightly pungent, soybean smell | Color Additive 2 | Yellow (0.4%) |
| KMFDS Approval | Functional ingredient with immune enhancement function | Color Additive 3 | Red (0.4%) |
| | | Color Additive 4 | Green (0.1%) |
| KMFDS Approval No. | No. 2007–15 | Flavor Additive 1 | Soybean paste (0.3%) |
| KMFDS Guidance | 1200 mg/day | Flavor Additive 2 | Grain powder (0.1%) |

throughout the process. Unblinding did not occur during the clinical trial. The code opening of the allocation schedule for statistical analysis occurred only after the blinded data review was complete. The overall clinical trial management and monitoring, database design and construction, data entry and validation, statistical analysis, and final report documentation were performed by an external contract research organization.

## Anthropometric parameters and blood collection

The body weight, fat percentage (UM0703581; Tanita, Tokyo, Japan) and height (GL-150; G-tech International, Uijeongbu, Korea) of the subjects were measured in the morning, with the subjects wearing lightweight clothes and no shoes; body mass index (BMI) was then calculated in units of kilograms per square meter ($kg/m^2$). Systolic and diastolic blood pressure (BP) was measured on the left arm using an automatic BP monitor (FT-200S; Jawon Medical, Gyeong-san, Korea) after a 20-min rest. The participants were instructed to not smoke or drink alcohol for at least 30 min before the BP measurement and fast for 12 h before the blood sample collection, which was conducted at the following check points: screening, week 0, week 4 and the final follow-up visit on week 8. The blood samples were also used to conduct routine checkups and detect any sign of acute conditions. Venous blood specimens were collected in EDTA-coated and plain tubes and were then centrifuged to yield plasma and serum, respectively, which were stored at -70°C until analysis.

## Preparation for and execution of the cytotoxicity assay

After the blood collection, to prepare for the isolation of the peripheral blood mononuclear cells (PBMCs) to be used as effector cells, the whole blood sample was immediately mixed with the same volume of RPMI 1640 medium (Gibco22400-089; Thermo Fisher Scientific, Waltham, Massachusetts, USA), gently overlaid on Histopaque® (1077; Sigma-Aldrich, Irvine, UK) and centrifuged for 20 min at 1800 rpm and 15°C. After the separation, the buffy coat layer was isolated, washed once with RPMI 1640 medium, and resuspended in 1 mL of 10% FBS medium (fetal bovine serum; Cat.#16000–044; Gibco, Thermo Fisher Scientific, Waltham, Massachusetts, USA) diluted with RPMI 1640. An automated cell counter (LUNA-II; Logos Biosystems, Anyang, Korea) was used to identify and count the live cells among the PBMCs, which completed the effector cell preparation for the assay. Human myeloid leukemia cells (K562; Korean Cell Line Bank, KCLB No. 10243; Korean Cell Line Research Foundation, Seoul, Korea) were used because these cells are highly sensitive and receptive *in vitro* target of NK cells. Bottled and frozen K562 was withdrawn from the liquid nitrogen tank and immediately defrosted in a 37°C water bath for 10 min. The defrosted K562 was transferred into a conical tube containing 10% FBS medium. The cells were centrifuged, collected, resuspended in fresh FBS medium and distributed into Erlenmeyer flasks ready for cultivation. K562 was cultured in an incubator at 37°C under 5% $CO_2$, and its medium was replaced every 3 days along with regular monitoring to detect any abnormality in morphological changes and cell spheroids. After confirming that K562 reached the log phase, which usually occurred within approximately one week with its peak density approximating 0.75 million/mL, the target cells were centrifuged and collected for the cytotoxicity assay. After completing the preparations of both the effector and target cells for the cytotoxicity assay, the isolated PBMCs (E) were seeded into 96-well plates with K562 cells (T) at E:T ratios of 50:1, 10:1, 5:1 and 2.5:1 and incubated at 37°C under 5% $CO_2$ for at least 4 h. The cytolytic activities of NK cells were analyzed *via* the CytoTox 96® Non-Radioactive Cytotoxicity Assay Kit (G1782; Promega Co., Fitchburg, Massachusetts, USA) according to the manufacturer's instructions. The color reactions were read at 490 nm using the Victor X5 2030 multilabel plate reader (Victor X5 2030–0050;

PerkinElmer, Hopkinton, Massachusetts, USA), and the results were calculated by the following formula:

$$\% \text{Cytotoxicity} = (\text{Experimental-Effector Spontaneous-Target Spontaneous})/$$
$$(\text{Target Maximum-Target Spontaneous}) \times 100$$

We adhered to the official guidelines published by KMFDS regarding evaluating NK cell cytotoxicity and the assay kit manufacturer's instructions regarding using colorimetric methodology. The assays were performed on-site on the same day as the blood collection under randomized and double-blinded conditions, and the obtained assay results were kept blinded until the code open. The PBMCs were used as a whole without the isolation of NK cells and discarded without storage after the analyses were completed.

## Biochemical and biomarker assays

Hematological analysis was performed using a HORIBA ABX automated blood counting analyzer (HORIBA ABX Micro ES60; HORIBA Ltd., Tokyo, Japan). Serum glucose concentrations were measured according to the hexokinase method on a Hitachi 7600 autoanalyzer (Hitachi 7600 Modular; Hitachi Group, Tokyo, Japan). Serum albumin concentrations were analyzed through the bromocresol green (BCG) method using an ALB kit (DCA Vantage Analyzer; SIEMENS, Tarrytown, New York, USA) with an ADVIA 2400 autoanalyzer (ADVIA 2400 Chemistry System; SIEMENS, Tarrytown, New York, USA). Leukocyte counts were determined using the HORIBA ABX diagnostic analyzer (HORIBA ABX Micro ES60; HORIBA Ltd., Tokyo, Japan). ALT and AST levels were measured by the International Federation of Clinical Chemistry and Laboratory Medicine (IFCC) UV method using a Hitachi 7600 automatic analyzer (Hitachi 7600 Modular; Hitachi Ltd, Tokyo, Japan). The URiSCAN system (URiSCAN 10 SGL Strip; YD Diagnostics, Yong-In, Korea) was used to evaluate urine specimens for the diagnosis of renal diseases and systemic adverse effects. The urine specimens were analyzed for specific gravity, pH, leukocyte count, nitrite, protein, glucose, ketones, urobilinogen, bilirubin, and occult blood. Immunoglobulin (Ig) G1, IgG2 and IgM were measured by the immune nontuberculous method using a Bep II instrument (BEP 2000 Advance System; SIEMENS, Tarrytown, New York, USA).

## Statistical analysis

For the general characteristics and survey results, noncontinuous variables are presented as numerical values and percentages, whereas continuous variables are presented as averages and means ± standard deviations or ± standard errors. Analyses were performed using a per-protocol approach. All data were analyzed using SAS® (Version 9.4; SAS Institute, Cary, North Carolina, USA). The Kolmogorov-Smirnov test was used to test normality. The Wilcoxon signed-rank test (for skewed variables) and independent *t*-tests (for normally distributed variables) were used to compare parameters between the control (placebo) and test (Bio-Germanium) groups. Chi-square tests were used for categorical variables. ANCOVA was used to adjust the baseline values for further comparison. Comparisons between the baseline measurements and the measurements collected at follow-up visits for each group were performed *via* paired *t*-tests. A two-tailed *P*-value of less than 0.05 was considered statistically significant.

## Results

Our clinical trial discovered novel findings for the mechanism of germanium immunostimulation in the human body by revealing the increased cytotoxicity of NK cells and activation of

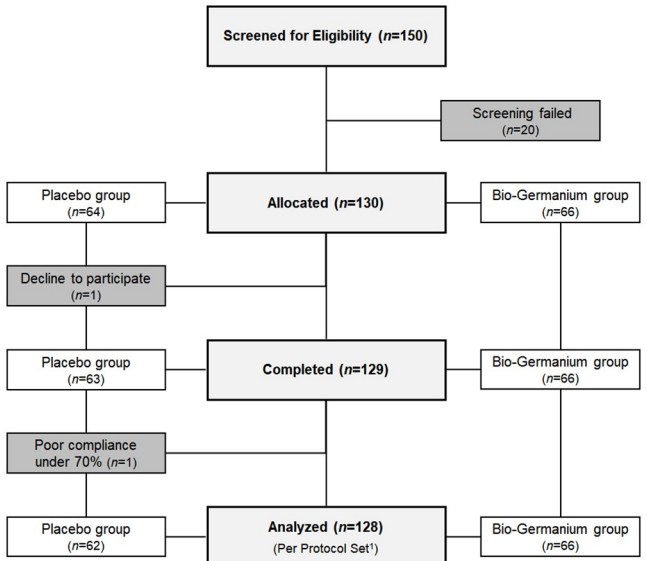

**Fig 1. Flow diagram of subject randomization.** [1] Completed the clinical trial without major protocol violations.

IgG1. The final results of this study included 128 subjects, excluding one subject who voluntarily dropped out and one who had poor compliance (less than 70%). No adverse events were observed during the clinical trial. The first participant was enrolled on January 9th, 2018, and the last (130th) participant's final visit ended on June 5th, 2018. A flow diagram of the randomization of subjects is presented in Fig 1.

## Basic characteristics of the subjects

Table 2 outlines the basic baseline characteristics of the two groups. At baseline, there were no significant differences between the two groups in age, sex distribution, weight, BP, exercise status, family history, and the ratio of smokers/alcohol drinkers. We used the leukocyte count as the subject inclusion criterion for normal immune function, and significant differences were not found at baseline between the two groups. The Bio-Germanium group recorded $5.38 \pm 1.06$ ($\times 10^3/\mu L$), while the placebo group recorded $5.52 \pm 1.02$ ($\times 10^3/\mu L$) as baseline leukocyte counts ($P > 0.05$).

## Natural killer cell cytotoxicity and immunoglobulin values

NK cell cytotoxic activities (%) were measured based on E:T ratios of 50:1, 10:1, 5:1 and 2.5:1. As outlined in Table 2, profound differences between the two groups were found in NK cell cytotoxicity at all E:T ratios (50:1 to 2.5:1) at follow-up. The NK cell cytotoxic activities in the Bio-Germanium group at E:T ratios of 50:1, 10:1, 5:1 and 2.5:1 were significantly higher than those in the placebo group at follow-up ($P < 0.05$) with respect to the baseline values (Table 3).

When we compared the changes in the two groups (Fig 2), the Bio-Germanium group exhibited NK cell cytotoxicity increases at E:T ratios of 50:1, 10:1, 5:1 and 2.5:1 ($12.60 \pm 32.91\%$, $10.19 \pm 23.88\%$, $9.28 \pm 16.49\%$ and $7.27 \pm 15.28\%$, respectively), but the placebo group showed decreases ($P < 0.01$). Notably, the Bio-Germanium group with an E:T ratio of 2.5:1 exhibited a near twofold increase from 8.91% at baseline to 16.19% at follow-up. The sub-analyses of the

**Table 2. Basic characteristics and anthropometric values of the subjects at baseline.**

| | Total Subjects (n = 128) | Placebo Group (n = 62) | Bio-Germanium Group (n = 66) | |
|---|---|---|---|---|
| | Baseline | Baseline | Baseline | P |
| Male / Female n, (%) | 13(10.16) / 115(89.84) | 4(6.45) / 58(93.55) | 9(13.64) / 57(86.36) | 0.179 |
| Age (year) | 53.64 ± 11.08 | 53.31 ± 10.67 | 53.95 ± 11.53 | 0.742 |
| Weight (kg) | 60.87 ± 9.40 | 60.86 ± 10.18 | 60.88 ± 8.68 | 0.989 |
| Height (cm) | 159.20 ± 6.21 | 158.85 ± 6.18 | 159.53 ± 6.26 | 0.539 |
| Leukocyte counts (×10³/μL) | 5.45 ± 1.04 | 5.52 ± 1.02 | 5.38 ± 1.06 | 0.427 |
| Systolic BP (mmHg) | 115.90 ± 12.32 | 115.53 ± 12.95 | 116.24 ± 11.78 | 0.746 |
| Diastolic BP (mmHg) | 70.48 ± 8.54 | 70.16 ± 8.96 | 70.77 ± 8.19 | 0.687 |
| Current smoker n, (%) | 3 (2.34) | 0 (0.00) | 3 (4.55) | 0.245[1] |
| Current drinker n, (%) | 43 (33.59) | 18 (29.03) | 25 (37.88) | 0.110 |
| Exercise +3 times per week n, (%) | 64 (50.00) | 29 (46.77) | 35 (53.03) | 0.771 |
| Family history of immune disease n, (%) | 0 (0.00) | 0 (0.00) | 0 (0.00) | - |

The data represent subject count, percentage and the mean ± standard deviation at baseline.

P-values were derived from a chi-square test (noncontinuous variables) or independent t-test (continuous variables).

[1] Derived from Fisher's exact test.

- Not applicable.

female subjects categorized into the pre- and post-menopause groups are provided in the S1 File.

For immunoglobulins, the difference in the IgG1 change from baseline to follow-up between the placebo group (-2079.03±5006.03 mg/dL) and the Bio-Germanium group (-275.76±4996.28 mg/dL) was statistically significant (P = 0.044, Table 4).

Although the changes from baseline to follow-up between the placebo and test groups for IgG2 (-1860.63±5286.78 mg/dL, -366.67±2953.09 mg/dL, respectively) and IgM (-0.13±9.44 mg/dL, 2.47±7.83 mg/dL, respectively) did not show statistically significant differences with P-values less than 0.05 (P = 0.076 for IgG2 and P = 0.092 for IgM), the trends in the IgG2 and

**Table 3. NK cell cytotoxicity values at baseline and follow-up.**

| (%) | Placebo Group (n = 62) | | | | | | Bio-Germanium Group (n = 66) | | | | | | $P^a$ | $P^b$ | $P^c$ |
|---|---|---|---|---|---|---|---|---|---|---|---|---|---|---|---|
| | Baseline | | | Follow-up | | | Baseline | | | Follow-up | | | | | |
| E:T = 50:1 | 49.44 | ± | 25.53 | 46.01 | ± | 24.57 | 43.27 | ± | 8.44 | 55.87 | ± | 27.85** | 0.125 | 0.011 | |
| Δ | | | | -3.44 | ± | 27.01 | | | | 12.60 | ± | 32.91 | | | 0.007 |
| E:T = 10:1 | 29.79 | ± | 17.42 | 24.67 | ± | 15.17* | 23.33 | ± | 20.54 | 33.52 | ± | 17.50*** | 0.006 | 0.002 | |
| Δ | | | | -5.12 | ± | 18.30 | | | | 10.19 | ± | 23.88 | | | <0.001 |
| E:T = 5:1 | 20.65 | ± | 13.66 | 14.06 | ± | 10.05** | 14.01 | ± | 10.86 | 23.29 | ± | 14.91*** | 0.003 | <0.001 | |
| Δ | | | | -6.59 | ± | 16.27 | | | | 9.28 | ± | 16.49 | | | <0.001 |
| E:T = 2.5:1 | 14.05 | ± | 16.19 | 9.20 | ± | 8.82* | 8.91 | ± | 10.49 | 16.19 | ± | 12.44*** | 0.020 | <0.001 | |
| Δ | | | | -4.86 | ± | 18.23 | | | | 7.27 | ± | 15.28 | | | <0.001 |

The data represent the mean ± standard deviation.

Δ represents the change from baseline at follow-up.

$P^a$-values were derived from the Wilcoxon rank-sum test at baseline between groups.

$P^b$-values were derived from the Wilcoxon rank-sum test at follow-up between groups.

$P^c$-values were derived from adjusted baseline ANCOVA for Δ between groups.

*P<0.05, **P<0.01 and ***P<0.001 values were derived from a paired t-test at follow-up from baseline within groups.

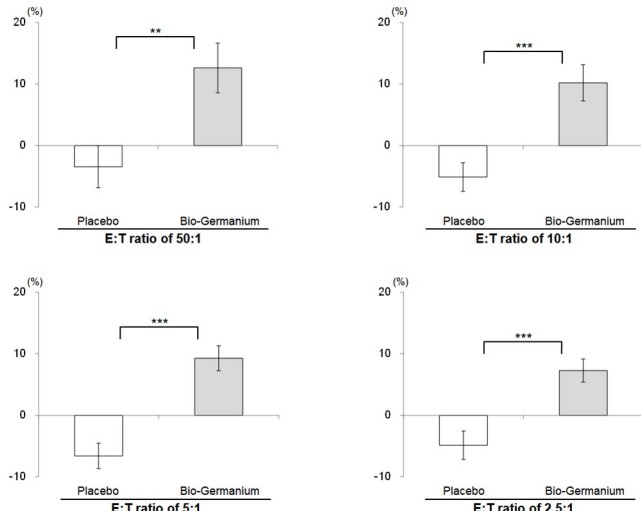

**Fig 2. Comparison of NK cell cytotoxicity Δ values before and after supplementation.** The data represent the mean ± standard error. Δ represents the change from baseline at follow-up. $**P<0.01$ and $***P<0.001$ values were derived from adjusted baseline ANCOVA for Δ between groups.

IgM variations were similar to that of IgG1 (Fig 3), with the Bio-Germanium group exhibiting a better performance than the placebo group.

## Anthropometric, biochemical and urinary analysis of the safety set

Based on the safety set with 130 subjects, anthropometric, biochemical and urinary analyses were performed under the intention-to-treat (ITT) principle in a set of subjects who received the product and completed at least one safety evaluation [64, 65]. The anthropometric and biochemical data revealed no abnormal findings (Tables 5 and 6). General characteristics of blood composition observed in white blood cells, red blood cells, hemoglobin, hematocrit, and

**Table 4. Immunoglobulin G1, G2 and M values at baseline and follow-up.**

| (mg/dL) | Placebo Group (n = 62) | | | | | | | Bio-Germanium Group (n = 66) | | | | | | | $P^a$ | $P^b$ | $P^c$ |
|---|---|---|---|---|---|---|---|---|---|---|---|---|---|---|---|---|---|
| | Baseline | | | | Follow-up | | | Baseline | | | | Follow-up | | | | | |
| IgG1 | 82804.84 | ± | 15720.12 | | 80725.81 | ± | 14306.39** | 79272.73 | ± | 15087.10 | | 78996.97 | ± | 16327.15 | 0.197 | 0.396 | |
| Δ | | | | | -2079.03 | ± | 5006.03 | | | | | -275.76 | ± | 4996.28 | | | *0.044* |
| IgG2 | 47364.52 | ± | 18764.35 | | 45503.89 | ± | 18729.89** | 43692.42 | ± | 18349.00 | | 43325.76 | ± | 17725.25 | 0.269 | 0.412 | |
| Δ | | | | | -1860.63 | ± | 5286.78 | | | | | -366.67 | ± | 2953.09 | | | *0.076* |
| IgM | 114.76 | ± | 63.12 | | 114.63 | ± | 63.27 | 109.68 | ± | 62.41 | | 112.15 | ± | 62.81* | 0.691 | 0.858 | |
| Δ | | | | | -0.13 | ± | 9.44 | | | | | 2.47 | ± | 7.83 | | | *0.092* |

The data represent the mean ± standard deviation.

Δ represents the change from baseline at follow-up.

$P^a$-values were derived from an independent *t*-test (IgG1, normally distributed) or the Wilcoxon rank-sum test (IgG2 and IgM, not normally distributed) at baseline between groups.

$P^b$-values were derived from the Wilcoxon rank-sum test at follow-up between groups.

$P^c$-values were derived from an independent *t*-test (IgG1 and IgM, normally distributed) or the Wilcoxon rank-sum test (IgG2, not normally distributed) for Δ between groups.

$*P<0.05$ and $**P<0.01$ values were derived from a paired *t*-test at follow-up from baseline within groups.

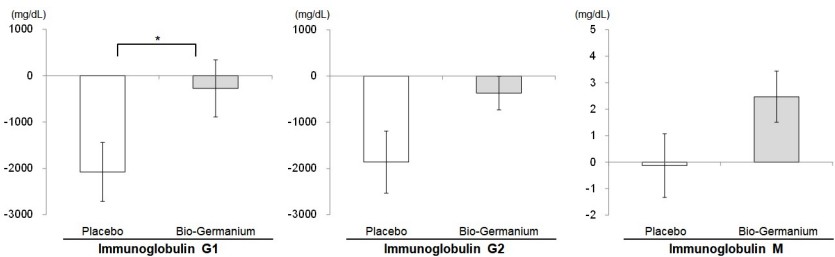

**Fig 3. Comparison of immunoglobulin G1, G2 and M Δ values before and after supplementation.** The data represent the mean ± standard error. Δ represents the change from baseline at follow-up. *$P<0.05$ value was derived from an independent $t$-test for Δ between groups.

platelets showed no significant differences after supplementation between the test and placebo groups. Additionally, none of the biochemical parameters, including AST, ALT, total protein, glucose, total cholesterol, BUN, creatinine, uric acid and others, showed significant differences.

In particular, no significant differences in the changes in liver- and kidney-related biochemical markers, such as AST, ALT, creatinine, BUN and uric acid, after supplementation were found between the test and placebo groups. In addition, the results of urinary analyses of all subjects in the Bio-Germanium group were normal (Table 7).

As adverse events were not observed during our clinical trial in conjunction with the supporting results from these anthropometric, biochemical and urinary analyses, the safety of Bio-Germanium for human consumption, which has been supported by earlier preclinical and clinical studies as well as by the over 20-year-long history of safe use, was also revalidated by our clinical trial.

## Discussion

The novel findings of this randomized, double-blind, placebo-controlled study indicated that the immunostimulation mechanism of Bio-Germanium is associated with the activation of NK cells and immunoglobulin.

Compared with the placebo group, the Bio-Germanium group showed significantly greater increases in NK cell activity at E:T ratios of 50:1, 10:1, 5:1, and 2.5:1. The immunostimulatory

**Table 5. Anthropometric data of the subjects (safety set).**

|  | Placebo Group ($n$ = 64) | | | Bio-Germanium Group ($n$ = 66) | | | |
|---|---|---|---|---|---|---|---|
|  | Baseline | Follow-up[1] | Δ | Baseline | Follow-up | Δ | P |
| Systolic BP (mmHg) | 115.53 ± 12.78 | 114.73 ± 12.69 | -0.71 ± 10.18 | 116.24 ± 11.78 | 116.35 ± 13.14 | 0.11 ± 11.47 | 0.669 |
| Diastolic BP (mmHg) | 70.20 ± 8.83 | 70.11 ± 8.99 | -0.03 ± 8.09 | 70.77 ± 8.19 | 71.06 ± 9.18 | 0.29 ± 8.32 | 0.825 |
| Heart Rate (bpm) | 77.41 ± 8.68 | 78.06 ± 9.44 | 0.75 ± 8.11 | 77.08 ± 10.58 | 75.77 ± 10.14 | -1.30 ± 8.71 | 0.170 |
| Body Temperature (°C) | 36.20 ± 0.16 | 36.13 ± 0.14 | -0.07 ± 0.19 | 36.20 ± 0.16 | 36.14 ± 0.14 | -0.05 ± 0.21 | 0.633 |
| Weight (kg) | 61.08 ± 10.17 | 60.71 ± 10.41 | -0.14 ± 1.13 | 60.88 ± 8.68 | 60.92 ± 8.72 | 0.04 ± 1.03 | 0.340 |

The data represent the mean ± standard deviation.

Δ represents the change from baseline at follow-up.

$P$-values were derived from an independent $t$-test for Δ between groups.

[1] Safety set analysis of the placebo group at follow-up consisted of 63 participants, excluding one subject's voluntary dropout.

**Table 6. Biochemical data of the subjects (safety set).**

| | Placebo Group (*n* = 64) | | | Bio-Germanium Group (*n* = 66) | | | |
|---|---|---|---|---|---|---|---|
| | Baseline | Follow-up[1] | Δ | Baseline | Follow-up | Δ | P |
| White Blood Cell (×10³/μL) | 5.53 ± 1.01 | 5.47 ± 1.39 | -0.05 ± 1.33 | 5.38 ± 1.06 | 5.17 ± 1.15 | -0.21 ± 1.22 | 0.483 |
| Red Blood Cell (×10⁶/μL) | 4.62 ± 0.66 | 4.72 ± 0.69 | 0.10 ± 0.85 | 4.62 ± 0.58 | 4.78 ± 0.77 | 0.15 ± 0.93 | 0.733 |
| Hemoglobin (g/dL) | 13.50 ± 1.36 | 13.70 ± 1.89 | 0.20 ± 1.72 | 13.69 ± 1.84 | 13.99 ± 2.25 | 0.30 ± 2.59 | 0.805 |
| Hematocrit (%) | 41.94 ± 6.13 | 42.63 ± 6.30 | 0.70 ± 7.86 | 41.96 ± 5.47 | 43.33 ± 7.47 | 1.37 ± 8.61 | 0.647 |
| Platelet (×10³/mm³) | 251.81 ± 63.92 | 225.79 ± 52.22 | -25.38 ± 52.96 | 239.17 ± 57.83 | 220.68 ± 50.76 | -18.48 ± 62.81 | 0.502 |
| Lymphocyte (×10³/mm³) | 1.96 ± 0.47 | 1.90 ± 0.51 | -0.05 ± 0.41 | 1.97 ± 0.54 | 1.86 ± 0.48 | -0.11 ± 0.36 | 0.413 |
| Monocyte (×10³/mm³) | 0.18 ± 0.07 | 0.17 ± 0.08 | -0.01 ± 0.09 | 0.19 ± 0.09 | 0.17 ± 0.07 | -0.02 ± 0.09 | 0.416 |
| AST (IU/L) | 23.52 ± 7.95 | 21.38 ± 6.96 | -1.57 ± 3.84 | 22.59 ± 6.52 | 21.12 ± 5.60 | -1.47 ± 4.17 | 0.886 |
| ALT (IU/L) | 20.31 ± 13.97 | 18.86 ± 13.01 | -0.70 ± 6.37 | 19.68 ± 9.09 | 18.48 ± 7.69 | -1.20 ± 4.68 | 0.615 |
| Total Protein (g/dL) | 7.34 ± 0.35 | 7.50 ± 0.33 | 0.17 ± 0.33 | 7.32 ± 0.36 | 7.48 ± 0.35 | 0.16 ± 0.31 | 0.913 |
| Glucose (mg/dL) | 90.89 ± 10.64 | 92.49 ± 11.67 | 1.68 ± 8.02 | 88.61 ± 10.18 | 91.82 ± 10.14 | 3.21 ± 7.00 | 0.250 |
| Total Cholesterol (mg/dL) | 194.86 ± 39.98 | 205.95 ± 42.67 | 10.13 ± 22.85 | 194.70 ± 38.31 | 201.68 ± 38.36 | 6.98 ± 18.24 | 0.389 |
| BUN (mg/dL) | 14.48 ± 3.78 | 14.44 ± 3.62 | -0.08 ± 3.30 | 14.53 ± 0.18 | 14.56 ± 3.44 | 0.03 ± 2.87 | 0.841 |
| Creatinine (mg/dL) | 0.69 ± 0.14 | 0.66 ± 0.12 | -0.03 ± 0.09 | 0.71 ± 0.16 | 0.69 ± 0.16 | -0.02 ± 0.10 | 0.576 |
| Uric acid (mg/dL) | 4.57 ± 1.20 | 4.61 ± 1.08 | 0.03 ± 0.70 | 4.48 ± 1.03 | 4.58 ± 1.09 | 0.10 ± 0.65 | 0.575 |
| Ca (mg/dL) | 9.36 ± 0.37 | 9.40 ± 0.33 | 0.03 ± 0.39 | 9.38 ± 0.35 | 9.39 ± 0.28 | 0.01 ± 0.33 | 0.778 |
| P (mg/dL) | 3.80 ± 0.47 | 3.84 ± 0.49 | 0.03 ± 0.45 | 3.77 ± 0.44 | 3.86 ± 0.43 | 0.09 ± 0.33 | 0.375 |

The data represent the mean ± standard deviation.

Δ represents the change from baseline at follow-up.

*P*-values were derived from an independent *t*-test for Δ between groups.

[1] Safety set analysis of the placebo group at follow-up consisted of 63 participants, excluding one subject's voluntary dropout.

capability found in our study shows that Bio-Germanium augmented NK cell activity, one of the key markers of immune strength [66–68]. Additionally, the change in IgG1 from the baseline level was significantly different between the Bio-Germanium group and the placebo group.

**Table 7. Urinary analysis data of the subjects (safety set).**

| | Placebo Group (*n* = 63)[1] | | Bio-Germanium Group (*n* = 66) | | |
|---|---|---|---|---|---|
| | Normal Results | Abnormal Findings | Normal Results | Abnormal Findings | P |
| Specific Gravity n, (%) | 63 (100.0) | 0 (0.0) | 66 (100.0) | 0 (0.0) | - |
| pH n, (%) | 63 (100.0) | 0 (0.0) | 66 (100.0) | 0 (0.0) | - |
| Protein n, (%) | 63 (100.0) | 0 (0.0) | 66 (100.0) | 0 (0.0) | - |
| Glucose n, (%) | 63 (100.0) | 0 (0.0) | 66 (100.0) | 0 (0.0) | - |
| Ketone n, (%) | 63 (100.0) | 0 (0.0) | 66 (100.0) | 0 (0.0) | - |
| Bilirubin n, (%) | 63 (100.0) | 0 (0.0) | 66 (100.0) | 0 (0.0) | - |
| Urobilinogen n, (%) | 63 (100.0) | 0 (0.0) | 66 (100.0) | 0 (0.0) | - |
| Nitrite n, (%) | 63 (100.0) | 0 (0.0) | 66 (100.0) | 0 (0.0) | - |
| Leukocyte n, (%) | 63 (100.0) | 0 (0.0) | 66 (100.0) | 0 (0.0) | - |
| Erythrocyte n, (%) | 63 (100.0) | 0 (0.0) | 66 (100.0) | 0 (0.0) | - |

The data represent results at follow-up.

[1] Safety set analysis of the placebo group at follow-up consisted of 63 participants, excluding one subject's voluntary dropout.

⁻ Not applicable.

This result is consistent with that of a previous clinical study showing the immunostimulatory effect of Bio-Germanium in humans. Lee *et al.* [38] reported that in fifty human subjects ranging in age from 50 to 75 years, B cell (CD19) activity and TNF-α production were increased in the Bio-Germanium group, as indicated by flow cytometric analysis using a monoclonal antibody and by TNF-α enzyme immunometric assay. Similarly, our findings are consistent with those of previous *in vivo* studies. Joo *et al.* [39] conducted a plaque-forming cell (PFC) assay and evaluated the effect on antibody production in mice fed Bio-Germanium. This study also showed that the proliferation of B cell subset compositions increased in a dose-dependent manner, from 48.9% (100 mg/kg) to 50.1% (200 mg/kg) to 53.2% (400 mg/kg) to 55.6% (800 mg/kg), and that the PFC count also increased. Another study by Baek *et al.* [2] showed that the administration of Bio-Germanium induced nitric oxide production and increased superoxide anion release, thereby increasing phagocytosis, macrophage activation and TNF-α production. Most importantly, NK cell-mediated cytotoxicity was also increased in a dose-dependent manner in their *in vivo* study, a result revalidated by our clinical study.

As NK cells are partly activated in response to macrophage-derived cytokines [69], our hypothesis to focus on the NK cell profile in this study largely originated from the macrophage activations observed in these *in vivo* studies. Moreover, the emerging importance of NK cells in maintaining general health as well as in defending against malignant diseases, as revealed in various studies, has shown that NK cells are one of the key components in the protective function of the immune system and that the cytotoxicity of NK cells is one of the key indicators of immune system activation. Fundamentally, NK cell effector function has been recognized as a primary contributor to innate immunity in viral, bacterial and parasitic infections and as a crucial regulator for the initiation of the adaptive immune response [70]. NK cells can also contribute to protective responses against a variety of inflammatory pathologies and even cancers [71–73]. Furthermore, the cytotoxic activity of NK cells was used as a cancer prognostic marker in an 11-year follow-up study to predict the probability of future development of cancer, revealing that medium and high cytotoxic activity could reduce the relative risk of cancer incidence [74]. Additionally, well-preserved NK cytotoxicity can be considered a marker of healthy aging, while low NK cytotoxicity is a predictor of increased morbidity and mortality due to infections [75]. Integrating the above findings with the previous studies of Bio-Germanium, we developed our hypothesis that the prevailing mechanism of the immunostimulatory function of germanium is based on NK cells. Subsequently, our clinical results substantiated that the immunostimulatory capability of germanium supplementation was associated with augmented NK cell cytotoxicity.

Generally, the immunostimulation property of organic germanium is deemed to be the foundation of its various therapeutic effects, such as anticancer, antiaging, antiviral and anti-inflammatory effects—areas in which immune function plays a key role. The immune function activation mechanism of organic germanium is established by enriching the oxygen supply [48, 76, 77], detoxifying heavy metal elements [48, 78–80], scavenging reactive oxygen species (ROS) [81–83] and increasing reduced glutathione (GSH) levels [81, 84]. El-Din [85] reported that organic germanium showed radical scavenging activity immediately after tumor inoculation and γ-irradiation exposure, resulting in the suppression of tumor growth and in the reduction of cellular injury from γ-irradiation-induced toxicity, which were attributable to the inhibition of the cascade reaction of membrane lipid peroxidation. This finding is important because oxidative stress and the overproduction of ROS have been associated with the pathogenesis of tumors and cancer [86]. Additionally, Pronai and Arimori [87] reported that superoxide scavenging activity in patients with certain immunological disorders was significantly lower than that in healthy controls. Interestingly, the scavenging function of organic germanium has been observed on several different types of ROS, such as hydrogen peroxide,

superoxide anion, and hydroxyl radicals [7, 88, 89]. As supplementation of antioxidants decreases the level of oxidative stress by their ROS-scavenging activities and thereby helps maintain GSH at high levels [90], the scavenging activities of organic germanium can also explain its role in increasing GSH levels. In fact, GSH plays an important role in enhancing immune function [91]. Increasing the GSH concentration in relevant tissues elicits an antitumor effect by stimulating immunity through the GSH pathway [92], while GSH depletion status can lead to decreased lymphocyte proliferation and impaired macrophage and T cell function [93]. Collectively, scavenging ROS and increasing GSH levels by organic germanium can be deemed the founding mechanism for its immunostimulation and immune enhancement.

Finally, to our knowledge, this study is the first to examine immunoglobulin levels after germanium intervention in human subjects. IgG1 is the major serum immunoglobulin and is principally responsible for the recognition, neutralization, and elimination of pathogens and toxic antigens [94], and the activation of NK cells increases the IgG1 level and facilitates the production of IgG2a antibodies in the *in vivo* setting [95]. As the affinity of an immunoglobulin for an antigen is utilized to enhance immune effector functions [96] and hypofucosylated IgG1 and IgG3 enhance the effector functions of NK cells [97], the discovery of IgG activation in relation to the activation of NK cells by Bio-Germanium in a clinical setting is considered a novel finding of this study. Additionally, Bio-Germanium's activation of B cells (CD19) in humans was confirmed earlier by Lee *et al.* [38], which was deemed appropriate as CD19 serves as a signaling intermediary for the stimulation of CD86 on B cells, effecting the subsequent involvement of CD86 in the regulation of the IgG1 levels [98].

Because the humoral immunity component is activated by the secretion of antibodies, such as IgG, and the cell-mediated immunity component is activated by antigen-specific cytotoxic lymphocytes, we can suggest further exploration to identify future opportunities in the application of Bio-Germanium in B cell and T cell combined therapies. However, as our current study did not focus on investigating the activities of antibodies, more research in this area is warranted in the future. Even though this randomized placebo-controlled study clearly showed that supplementation with Bio-Germanium conferred the immunostimulatory effect, the primary focus of this study was on investigating the working mechanism behind Bio-Germanium's immunostimulation, not on investigating the specificity of effect on a particular disease. To initiate its drug discovery process, further investigation is required in areas such as disease-specific mechanisms and patient-specific effects.

We have also noticed that certain differences in findings from various studies exist due to dissimilarities between experimental tests and clinical trials, including subject selection. Our subjects were healthy adult volunteers between 20 and 75 years of age with normal immune function, while most previous studies were conducted with cancer patients or in *in vivo* settings; thus, the subjects' physiological and general environmental conditions should also be considered. In particular, some immune parameters were decreased at follow-up in both the control and test groups. Our research was carried out from January 9th to June 5th, and as most subjects were tested during the winter and spring, the immune-related parameters were possibly affected by the cold, windy weather in the winter [99] and Asian Dust (AD) outbreaks during the spring in Seoul. Regarding sudden temperature changes, Brenner et al. reported that cold exposure can contribute to deteriorations in immune function, including a reduction in the NK cell count [100]. In Seoul, the daily average temperature in January and February was below 0˚C (32˚F), and the lowest temperature was -18˚C (0˚F). The temperature abruptly increased to 22˚C (72˚F) in mid-March, decreased again to 0˚C (32˚F) in early April, and then surged to 26˚C (79˚F) in mid-April [101]. Such abrupt undulation of temperature within a short duration could be regarded quite extraordinary. In addition, AD, originating from the

deserts and drylands of Mongolia, affects Korea from March to May. One study noted that AD particles can alter immune response profiles [102]. The severe weather conditions and AD in Seoul during the trial period may have had a certain impact on our baseline differences and the declining trends observed in certain parameters in both the placebo and test groups.

Last but not least, if our study had not been bound by time and resource constraints, more immediate and frequent observations in the earlier stage of the immunocytokine changes, cytokine secretion assays in response to certain stimuli in selected immune cells from isolated PBMCs and additional exhaustive observations of NK cells during their isolation from PBMCs would have allowed us to capture a more holistic picture of the immunostimulation mechanism of Bio-Germanium as immunocytokine changes are sometimes instantaneous and short-lived. Also, to better understand the influence of Bio-Germanium on the antibody immune response, further evaluation of the activities of antigen-specific cytotoxic lymphocytes and the responses of antibodies to certain proteins and polysaccharide antigens are imperative to elucidate its mechanism in humoral immunity. Additionally, the use of clinical data from a single center, the imbalance in the male and female proportion, and the disproportional age distribution could be limitations, and caution is needed in generalizing these results to a broader population.

In conclusion, this clinical trial confirms that Bio-Germanium supplementation stimulates immune function by increasing the cytotoxicity of NK cells and activating immunoglobulin, in addition to activating B cells and TNF-$\alpha$, as observed in an earlier clinical study [38]. We have added newly discovered clinical findings for germanium's immunostimulation mechanism substantiated through a large-scale trial offering a universal basis by focusing on healthy human subjects, and have contributed to bridging the gap between preclinical and clinical results. We believe Bio-Germanium is a highly promising therapeutic agent and should certainly be further explored for immunotherapy where its immunostimulation effect can have potential development opportunities.

## Supporting information

**S1 File.**
(PDF)

**S2 File.**
(PDF)

**S3 File.**
(DOC)

**S1 Checklist. CONSORT 2010 checklist of information to include when reporting a randomised trial**\*.
(DOC)

## Author Contributions

**Conceptualization:** Jisuk Chae, Sa Rang Jeong, Dong Yeob Shin, Jong Ho Lee.

**Data curation:** Jung Min Cho, Sa Rang Jeong, Jong Ho Lee.

**Funding acquisition:** Jong Ho Lee.

**Investigation:** Min Jung Moon, Jong Ho Lee.

**Methodology:** Jung Min Cho, Jisuk Chae, Min Jung Moon, Jong Ho Lee.

**Project administration:** Jung Min Cho, Dong Yeob Shin.

**Resources:** Min Jung Moon.

**Software:** Jong Ho Lee.

**Supervision:** Dong Yeob Shin, Jong Ho Lee.

**Validation:** Jung Min Cho, Min Jung Moon, Dong Yeob Shin.

**Writing – original draft:** Jung Min Cho, Jong Ho Lee.

**Writing – review & editing:** Jung Min Cho, Jisuk Chae, Sa Rang Jeong, Min Jung Moon, Dong Yeob Shin, Jong Ho Lee.

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
