## [Decision Letter · Decision Letter 0]

20 Jul 2020

PONE-D-20-05271

Immune activation of Bio-Germanium in a randomized, double-blind, placebo-controlled clinical trial with 130 human subjects: therapeutic opportunities from new insights

PLOS ONE

Dear Dr. Lee,

Thank you for submitting your manuscript to PLOS ONE. After careful consideration, we feel that it has merit but does not fully meet PLOS ONE’s publication criteria as it currently stands. Therefore, we invite you to submit a revised version of the manuscript that addresses the points raised during the review process.

The reviewers have raised a number of concerns that need attention. They request additional information on methodological aspects of the study, including the sample size calculation, the suitability of the outcomes to make inferences about immune function and the assays used. Two reviewers have also commented on the gender imbalance of the participants in the study. In your revised manuscript please discuss how this may have influenced your results.

Could you please revise the manuscript to carefully address the concerns raised?

We look forward to receiving your revised manuscript.

Kind regards,

George Vousden

Senior Editor

PLOS ONE

Journal Requirements:

2. Thank you for submitting your clinical trial to PLOS ONE and for providing the name of the registry and the registration number. The information in the registry entry suggests that your trial was registered after patient recruitment began.

PLOS ONE strongly encourages authors to register all trials before recruiting the first participant in a study.

i) your reasons for your delay in registering this study (after enrolment of participants started);

ii) confirmation that all related trials are registered by stating: “The authors confirm that all ongoing and related trials for this drug/intervention are registered”.

Please also ensure you report the date at which the ethics committee approved the study as well as the complete date range for patient recruitment and follow-up in the Methods section of your manuscript.

'This work was supported by the New Drug Discovery Fund of Geranti Pharmaceutical.

The funders had no role in study design, data collection and analysis, decision to publish, or preparation of the manuscript.'

We note that you received funding from a commercial source: Geranti Pharmaceutical

Reviewers' comments:

Reviewer's Responses to Questions

**Comments to the Author**

1. Is the manuscript technically sound, and do the data support the conclusions?

Reviewer #1: Yes

Reviewer #2: No

Reviewer #3: Yes

Reviewer #4: Yes

2. Has the statistical analysis been performed appropriately and rigorously? 

Reviewer #1: No

Reviewer #2: Yes

Reviewer #3: Yes

Reviewer #4: Yes

3. Have the authors made all data underlying the findings in their manuscript fully available?

Reviewer #1: Yes

Reviewer #2: Yes

Reviewer #3: Yes

Reviewer #4: No

4. Is the manuscript presented in an intelligible fashion and written in standard English?

Reviewer #1: Yes

Reviewer #2: Yes

Reviewer #3: Yes

Reviewer #4: Yes

5. Review Comments to the Author

Reviewer #1: The sample size calculation doesn’t match the assumed effect size 5.76. The sample size 55/group will have 80% power to detect an effect size of 0.5 at a two-sided with a significance level (alpha) of 0.050 using a two-sided two-sample equal-variance t-test. If you wanted to test NK cell cytotoxic activity at an effector cell:target cell (E:T) ratio of 10:1 between -0.28±10.7% (mean ± standard deviation) in the placebo group and that in the test group 7.4±10.0%, you only need 30 patients each group. Please provide clear description about the sample size estimate.

Table 3, Pc-values should not be derived from adjusted baseline ANCOVA for Δ between groups. Please double check. They should be derived from an independent t-test or the Wilcoxon rank-sum test for Δ between groups.

Reviewer #2: The manuscript "Immune activation of Bio-Germanium in a randomized, double-blind, placebo-controlled clinical trial with 130 human subjects: therapeutic opportunities from new insights" is interesting, focusing on an interesting point of immune response regulation by organic salts of germanium. There are several articles studying this point, but as the authors report, just a few clinical trials in humans. The idea of studying the natural killer activity as well as humoral immune response is good, but there are some major drawbacks in the study that compromises the results.

1) The casuistry is composed of 128 healthy individuals, being 13 males and 115 females. This accounts for a significant difference between the sexes. The females' group were extremely larger than males' group (90% x 10%). It is well known that immune functions differs between sexes, so their particular data might have been discriminated.

2) Moreover, as the ages of study groups' were between 20 and 75 years old, it is fundamental to look for reproductive age in females (pre-menopausal x post-menopausal) to evaluate the immunological data (inflammation, NK cytotoxic activity, IgG subclass distribution etc.), as immune function is dependent on hormonal function as well. The immunoendocrine axis is well described nowadays, and immunocytes present receptors for hormones, as well as endocrine system cells respond to immune system agonists and antagonists. so, the data of female subjects might be evaluated according to age and sexual hormonal function.

3) Another major fault of the study is dependent on the evaluation of humoral immune response. The mere quantification of the immunoglobulin isotypes does not reflect the functionality of antibody responses. It is important to evaluate specific antibody responses to proteins and polysaccharide antigens to evaluate humoral immunity. It is well known that differentiation of B cells leading to the production of diverse isotypes of immunoglobulins depend on the exposition of B cells to different types of antigens in different ecological niches of the immune system (mucosae x systemic immunity, just to cite one example), under different microenvironments, different cytokines etc.

Therefore a wider evaluation of humoral immunity is of paramount importance to understand the influence of germanium organic salts in the specific antibody immune response.

4) In the discussion (lines 553-556), the authors tell that the affinity of an antibody to "its antigen" is important to enhance effector functions of NK cells in an antibody dependent cell cytotoxicity (ADCC) mechanism, the observation of increased levels of IgG1 could be directly related to NK stimulation. In my point of view, this conclusion cannot be drawn from your data, as the mere increase of IgG1 do not correlate to the possible activation of NK cells by ADCC.

There are some points that deserve more discussion. For example, why the placebo group immune function decreased from beginning to ending of experimental period? These alterations are noted in NK cell activity, as well as in IgG1 quantification.

Minor points: The abbreviations presented mainly in page 7 (lines 198-212) must be described before being presented in the text.

There are several incomplete references, such as ref 21-24, 85-87 and 94-96.

Reviewer #3: The article by Lee et al. addresses an important and timely topic, which is to examine changes in immune profiles following Bio-germanium supplementation in a randomized, double-blind, placebo-controlled study.

In general, the manuscript is very well written, presenting the data clearly and concisely. However, I have some comments, as follow:

1. The Abstract and the Introduction.

The topics were well written.

- In Line 53, I suggest adding in the sentence that the studies cited were performed in animal model or in cell lines culture.

- I suggest adding a reference in the sentence from line 90 to 95, or the number of registries in the agencies.

2. Methodology.

- In Line 240 I believe you would to say: “The fasting whole blood leukocytes…”, right? Correct please.

- Regarding the collection of the blood samples, were they collected only before the treatment and after 8 weeks?

-My main doubt in methodology is about the cytotoxicity of NK cells assay. As this test is not performed usually in routine of clinic laboratory, I suggest detailing the technic performed in this assay.

Regarding NK cells assay:

- Was all the NK assay performed in the same day? Were the assays performed in the same day of blood collection? Were the assays for Placebo and Germanium individuals carried out at the same time (doing a retroactive search after finding out which sample was from which group because of the blind study)?

- If the tests have not been carried out on the same day, detail the PBMC storage procedure used before the analysis.

- The PBMC were added in 96-well plates with K-562 cells. In PBMC, in addition to NK cells, there are different types of lymphocytes and monocytes. Were the monocytes removed prior join to K-562 cells? It seems to me that it would be very interesting to know the proportion of NK cells in the PBMC of each individual. This increase in cytotoxicity could be due to a change in the absolute number of NK cells or maybe effect of other cells, no? Comment.

- Even in very careful separations, sometimes residual red blood cells may remain in PBMC. As the assay is based on LDH activity and red blood cells have a lot of LDH, was this considered?

- Concerning K-562 cells, are there some special preparation before use? Is it important the K-562 cells are in log phase before use in the NK-cells assay? Describe in methodology.

- Why this E:T ratio were preferred? Why not an intermediary E:T ratio, like 20, 25 or 30:1?

3. Results.

In general, the results are shown in a clear way. But I have some comments:

- Why the authors presented some data as mean ± standard deviation and the main results (NK cells and immunoglobulins) in tables and figures as mean ± standard error? Do you think this is the most suitable way? It seems to me that using standard error is not the correct way to present the data. Can you please comment on that?

- Have you any hypothesis to the decreased observed in cytotoxicity and immunoglobulins in placebo group?

4.Discussion.

- The discussion is ok. The limitations are those appointed in the methodology, regarding NK cells assay. I am not sure if is possible to be so emphatic in your statements.

- I suggest remove finding of previous study in the last paragraph and keep the conclusion of this study only.

Reviewer #4: In general, the manuscript is well-written, and presents interesting results from what looks to have been a well-run study.

1) Please comment on the large sex imbalance in the sample. Only about 10% were male. This is presumably a result of the recruitment method(s) used.

Although the sample size is limited, the authors should perform a subset analysis for males and females separately. (An alternative is to perform analysis with sex as a main effect and as an interaction term with treatment in the ANCOVA approach.) This can be reported in a supplemental table, and summarized quite briefly in the manuscript.

Please add some remarks about the potential effects of this study imbalance to the discussion of the limitations of the study.

2) Corn starch was used as placebo. Although it seems likely that this would have no immune effects, this should probably be justified in the manuscript.

3) In general, although not strictly necessary, the readability of the tables would probably benefit from rounding at one rather than at two decimal places. The authors may wish to try this out and see.

4) In Table 3, there appear to be clear differences between the groups at baseline. Do the authors have any thoughts about why this might be? Could some other covariate potentially explain this difference?

5) Although nothing needs to be done in the presentation, note that the ANCOVA approach to the "delta" values requires some care in interpretation. That is, the results hold conditional on a given baseline level.

6) To help interpret the results in the paper, please include some supplemental figures that display the raw data as "spaghetti" plots. That is, each endpoint is displayed as a set of pre- and post- measurements connected by lines. Treatment groups are separated and juxtaposed.

7) In the results section (e.g., Line 408) please report actual p-values rather than "P<0.05").

8) As displayed in Figures 2 and 3, and visible in the manuscript, placebo seems rather active. Can the authors shed some light on this? Could there be a regression to the mean effect at play?

Especially in Figure 3, we see that the treatment group essentially shows no change, while the placebo group does show change. It seems very difficult to imply an effect to bio-germanium that seems to only exist in contrast to a marked placebo change from pre to post.

9) The data do not appear to be available to perform spot checks or to make more informed suggestions.

In addition, here are some minor suggested edits:

Line 80: Change to "clinical study to-date with a group of 130".

Line 149: Change to "from yeast by dissolution in either".

Line 171: Change to "demonstrated efficacy in areas".

Lines 175 and 192: Delete the decorative P-values here in the background section.

Lines 227 and 229: Change all occurrences of "a history" to "any history".

Line 236: Identify the confidence level associated with "1.23-3.88".

6. PLOS authors have the option to publish the peer review history of their article (what does this mean?). If published, this will include your full peer review and any attached files.

Reviewer #1: No

Reviewer #2: **Yes: **Dewton de Moraes-Vasconcelos

Reviewer #3: No

Reviewer #4: No

---

## [Author Response · Author response to Decision Letter 0]

26 Aug 2020

Dear Reviewer,

We are grateful for your review of our manuscript. In response to your comments, we revised the manuscript accordingly with explanation for your review. (Attached file; Response to reviewers)

Thank you again for the opportunity to revise our manuscript, and we hope that our revision meets your expectation.

Respectfully,

Jong Ho Lee, Ph.D.

---

## [Decision Letter · Decision Letter 1]

25 Sep 2020

Immune activation of Bio-Germanium in a randomized, double-blind, placebo-controlled clinical trial with 130 human subjects: therapeutic opportunities from new insights

PONE-D-20-05271R1

Dear Dr. Lee,

We’re pleased to inform you that your manuscript has been judged scientifically suitable for publication and will be formally accepted for publication once it meets all outstanding technical requirements.

Kind regards,

Stephen L Atkin, MD

Academic Editor

PLOS ONE

Additional Editor Comments (optional):

Reviewers' comments:

Reviewer's Responses to Questions

**Comments to the Author**

1. If the authors have adequately addressed your comments raised in a previous round of review and you feel that this manuscript is now acceptable for publication, you may indicate that here to bypass the “Comments to the Author” section, enter your conflict of interest statement in the “Confidential to Editor” section, and submit your "Accept" recommendation.

Reviewer #1: All comments have been addressed

Reviewer #2: All comments have been addressed

Reviewer #3: All comments have been addressed

Reviewer #4: All comments have been addressed

2. Is the manuscript technically sound, and do the data support the conclusions?

Reviewer #1: (No Response)

Reviewer #2: Yes

Reviewer #3: Yes

Reviewer #4: (No Response)

3. Has the statistical analysis been performed appropriately and rigorously? 

Reviewer #1: (No Response)

Reviewer #2: Yes

Reviewer #3: Yes

Reviewer #4: (No Response)

4. Have the authors made all data underlying the findings in their manuscript fully available?

Reviewer #1: (No Response)

Reviewer #2: Yes

Reviewer #3: Yes

Reviewer #4: (No Response)

5. Is the manuscript presented in an intelligible fashion and written in standard English?

Reviewer #1: (No Response)

Reviewer #2: Yes

Reviewer #3: Yes

Reviewer #4: (No Response)

6. Review Comments to the Author

Reviewer #1: (No Response)

Reviewer #2: In my póint of view the manuscript is now ready to be accepted for publication. The authors revised the manuscript thoroughly and answered adequately to the question presented by the reviewers.

Reviewer #3: After re-evaluating the above manuscript, I consider the editions made and all the answers to previous questions very satisfactory. I congratulate the authors for responding with good arguments to all the questions asked.

Reviewer #4: The authors have not only provided a thorough and comprehensive response, but have also conducted additional analysis and evaluation of the data, as well as making further improvements in the presentation. Every point was dealt with carefully.

The level of academic rigor shown is quite impressive and frankly, a joy to see. This reviewer very much appreciated the attention to detail and the care taken both in the initial statistical analysis and in the response. While this reviewer's opinion is of very little note, it is yet very encouraging to see that the investigators hold themselves to an extremely high standard in their scholarship, and offers hope that these values will be passed to the next generation.

7. PLOS authors have the option to publish the peer review history of their article (what does this mean?). If published, this will include your full peer review and any attached files.

Reviewer #1: No

Reviewer #2: **Yes: **Dewton de Moraes Vasconcelos

Reviewer #3: No

Reviewer #4: No

---

## [Editor Report · Acceptance letter]

29 Sep 2020

PONE-D-20-05271R1 

Immune activation of Bio-Germanium in a randomized, double-blind, placebo-controlled clinical trial with 130 human subjects: therapeutic opportunities from new insights 

Dear Dr. Lee:

I'm pleased to inform you that your manuscript has been deemed suitable for publication in PLOS ONE. Congratulations! Your manuscript is now with our production department. 

Kind regards, 

on behalf of

Dr. Stephen L Atkin 

Academic Editor

PLOS ONE